# Frequency of Vitamin D Receptor Gene Polymorphisms in a Population with a very High Prevalence of Vitamin D Deficiency, Obesity, Diabetes and Hypertension

**DOI:** 10.3390/biomedicines11041202

**Published:** 2023-04-18

**Authors:** Salah Gariballa, Ghada S. M. Al-Bluwi, Javed Yasin

**Affiliations:** Internal Medicine, College of Medicine and Health Sciences, United Arab Emirates University, Al-Ain P.O. Box 15551, United Arab Emirates; ghadabluwi@uaeu.ac.ae (G.S.M.A.-B.); javed.yasin@uaeu.ac.ae (J.Y.)

**Keywords:** vitamin D, VDR receptor, obesity, diabetes, hypertension

## Abstract

Background: Although vitamin D levels and underlying vitamin D receptor (VDR) genetic polymorphisms have been linked to many common diseases including obesity, the association remains unclear. There is also co-existence of pathologically high proportions of obesity and vitamin D deficiency conditions in our UAE society. We therefore aimed to determine the genotypes and allele percentage frequency distribution of four polymorphisms—FokI, BsmI, ApaI and TaqI—in the VDR gene in healthy Emirati individuals and their association with vitamin D levels and chronic conditions including diabetes mellitus, hypertension and obesity. Methods: 277 participants who were part of a randomized controlled trial had their assessment that included clinical and anthropometric data. Whole blood samples were taken for measurements of vitamin D [25(OH) D], four vitamin D receptor gene polymorphism SNPs, including BsmI, FokI, TaqI and ApaI, metabolic and inflammatory markers and related biochemical variables. Multiple logistic regression analysis was used to assess the influence of vitamin D receptor gene SNPs on vitamin D status after adjusting for clinical parameters known to influence vitamin D status in the study population. Results: Overall, 277 participants with a mean (±SD) age of 41 ± 12, 204 (74%) of them being female, were included in the study. There were statistically significant differences in vitamin D concentrations between different genotypes of the four VDR gene polymorphisms (*p* < 0.05). There were, however, no statistically significant differences in vitamin D concentrations between subjects with and those without the four VDR gene polymorphisms genotype and alleles except for AA and AG and allele G in Apal SNP (*p* < 0.05). Multivariate analysis revealed no significant independent associations between vitamin D status and the four VDR gene polymorphisms after adjusting for dietary intake, physical activity, sun exposure, smoking and body mass index. In addition, no significant differences were found in the frequency of the genotypes and alleles of the four VDR genes among patients with obesity, diabetes and hypertension compared to those without these medical conditions. Conclusions: Although we found statistically significant differences in vitamin concentrations between different genotypes of the four VDR gene polymorphisms, multivariate analysis revealed no association after adjusting for clinical parameters known to influence vitamin D status. Furthermore, no association was found between obesity and related pathologies and the four VDR gene polymorphisms.

## 1. Introduction

Vitamin D an active hormone is essential for both bone and muscle health. There is also mounting evidence that vitamin D can convey other health benefits because of the discovery that most cells and tissues in the human body contain vitamin D receptors [1]. Vitamin D elicits its biological action through binding to the vitamin D receptor (VDR) [1]. A number of loss-of-function mutations of the VDR gene have been identified which result in hereditary vitamin-D-resistant rickets [2]. In addition, several subtle allelic polymorphisms which have been linked to diseases such as osteoporosis and metabolic bone disorders have been identified in the VDR gene [3]. However, the health impacts of these single-nucleotide polymorphisms (SNPs) are not well understood [4]. More SNPs are expected to be identified in future because of the large size of the VDR gene and a number of those already described have been found to be associated with some chronic diseases, including multiple sclerosis, rheumatoid arthritis, and some autoimmune and metabolic diseases [5,6,7,8]. In addition, the relationship between vitamin D and its VDR receptor polymorphism to obesity and related pathologies have been described [9,10]. Both obesity and vitamin D deficiency are highly prevalent in UAE citizens; however, their adverse and additive health implications are not yet clear. Recent evidence points to a role of vitamin D deficiency as a mediator for increased inflammation associated with obesity and related diabetes [11]. Furthermore, several observational studies have raised the possibility of the role of vitamin D in the development of type 2 diabetes and noted that low plasma 25(OH) D concentrations might be a mediator between obesity and increased risk of diabetes [12]. Furthermore, recent evidence also shows that vitamin D metabolism, storage, and action are affected by adiposity and that obese individuals need higher vitamin D doses compared to normal-weight individuals [13].

The BsmI, FokI, TaqI and ApaI single-nucleotide polymorphisms in the VDR gene are the most common and frequently studied [14,15]. Although numerous studies have reported the frequency of VDR gene SNPs and their associations with different diseases in different ethnic groups (mostly Caucasian populations), very few studies have been conducted in the Middle East, namely the UAE. Moreover, many of those studies conducted in the Middle East suffer from methodological flaws, including selection bias and small sample sizes [13,14,15,16,17]. Mutations of the VDR gene and allelic polymorphisms may influence disease susceptibility and responses to levels of circulating vitamin D, especially in areas where D deficiency is highly prevalent, such as in Asian and Middle Eastern populations. However, results of the studies which have been published so far on the relationship between low vitamin D status and its health implications in non- Caucasian populations yielded conflicting results [18,19]. For example, a cross-sectional study evaluated whether Pakistanis living in the city of Oslo have increased bone turnover compared with ethnic Norwegians due to their high prevalence of vitamin D deficiency [18]. The results revealed only minor ethnic differences in bone turnover, despite a striking difference in prevalence of secondary hyperparathyroidism with no differences observed in bone mineral density between the two groups [18]. Another study from the UK on vitamin D status and markers of bone turnover in Caucasian and South Asian postmenopausal women found that although South Asian women had significantly higher serum parathyroid hormones and lower 25(OH) D concentrations, there were no significant differences between the two groups for biochemical markers of bone turnover [19]. An interesting speculative explanation for this paradox is that altered metabolism of vitamin D due to ethnic and/or genetic differences in the Asian women may protect their skeletons from bone loss [20,21,22]. This is clearly an area for research. In summary, vitamin D deficiency and obesity and related pathologies are highly prevalent in UAE citizens, but their combined health implications are not yet clear. Mutations of the VDR gene and the presence of SNPs may account for variation in individual responses to levels of circulating vitamin D. The overall objective of this study is to investigate the genotypes and allele percentage frequency distribution of four common polymorphisms, FokI, BsmI, ApaI and TaqI, in the VDR gene in healthy Emirati individuals and their association with Vitamin D levels and chronic conditions, including diabetes mellitus, hypertension and obesity. 

## 2. Materials and Methods

Details of the subjects’ recruitment and methods have been published previously [23]. Briefly, healthy community free-living Emirati (UAE citizens) and expatriate Arabian men and women from other countries in the Middle East who were part of a randomized, double-blind placebo-controlled placebo intervention trial were included in this study. Apparently healthy subjects were recruited by local advertisement, from community health centers and from hospital out-patients in Al Ain city, which has two main teaching hospitals serving a total population of 400,000. Individuals with renal disease or stones, hypercalcemia, on calcium and/or vitamin D supplementation, bisphosphonates, steroid medications, hormones or diuretics or unable to give informed written consent were excluded. Following informed written consent, the eligible subjects’ blood samples were taken for measurements of vitamin D, vitamin D receptor gene polymorphism SNPs and inflammatory and metabolic risk markers. Clinical assessment that included demographic and baseline characteristics, general and self-rated health and physical activity were performed at baseline. Information on other important variables likely to influence vitamin D status was also collected. Al Ain Medical District Human research ethics committee approved the study protocol and consent to participate form (Reference number: AAMDHREC protocol no 14/15). Written consent obtained from all patients recruited to this study.

### 2.1. The Measurements

Face-to-face questionnaire data were collected on lifestyle and health factors that are of interest in this study of the health implications of vitamin D deficiency in UAE citizens. A common set of questions on education and socio-economic status, current and past occupation, history of previous illness or surgical operations, tobacco smoking, consumption of beverages, physical activity, use of herbal medicine, vitamin supplements and exogenous hormones for contraception and postmenopausal replacement therapy were collected. Anthropometric data including body weight, height and body mass index (BMI) were measured using the Tanita body composition analyzer. Using WHO sex-adjusted cut-of-points for BMI, subjects with BMI = 18–25 were classified as normal, BMI = 25.1–29.9 as overweight and those with BMI 30 as obese. 

### 2.2. Biochemical and Urine Analysis

Patients provided a fasting morning blood sample. Biochemical analysis of 25(OH) D was measured using a fully automated COBAS e411 analyzer that uses a patented Electro Chemiluminescence (ECL) technology for immunoassay analysis from ROCHE diagnostics, Mannheim, Germany.

### 2.3. DNA Preparation and VDR SNPs Genotyping Analysis

Genomic DNA was extracted from whole blood collected from study participants using a QIAamp DNA Mini Kit (QIAGEN, Valencia, CA, USA) according to the manufacturer’s instructions. The extracted genomic DNA was analyzed using agarose gel electrophoresis, quantitatively determined by spectrophotometry, and stored at 80 °C until use. The four VDR SNPs (BsmI, FokI, TaqI and ApaI) were evaluated using the TaqMan SNP genotyping assay, which consists of a predesigned mix of unlabeled polymerase chain reaction (PCR) primers and the TaqMan^®^ minor groove binding group (MGB) probe (FAM™ and VIC^®^ dye-labeled). All TaqMan SNP genotyping assays are designed to work with the TaqMan^®^ Universal PCR Master Mix, which contains DNA polymerase, dNTPs and optimized mix components and uses the same thermal conditions (Applied Biosystems, Foster City, CA, USA). The PCR consisted of a hot start at 95 °C for 10 min followed by 40 cycles of 94 °C for 15 s and 60 °C for 1 min. Fluorescence detection was performed at 60 °C. All assays were performed in 10–25 μL reactions, using TaqMan Genotyping Master Mix on 96-well plates using Genetic Analyzer (Applied Biosystems, Foster City, CA, USA) according to the manufacturer’s instructions.

### 2.4. Statistical Power of the Study

The sample size of 277 would allow the detection of a true mean difference in plasma 25(OH) D of 2.0 (ng/mL) [given the within group SD 8 (ng/mL)] between subjects with and without the VDR gene polymorphism genotype. This sample size would also allow the detection of a 10% difference in the genotypes or allele frequency distribution between subjects with and without obesity (assuming the prevalence is 5% in non-obese subjects and 15% in obese subjects) with 80% power and a type 1 error probability of ≤0.05.

### 2.5. Statistical Analysis 

Statistical analysis was performed with SPSS software, version 25.0 (SPSS Inc., Chicago, IL, USA). One-way ANOVA or the nonparametric Kruskal-Wallis H test was used to test for between-group differences, and a *p* value < 0.05 was considered significant. Multiple regression analyses were used to assess the independent association between vitamin D status (deficiency vs. insufficiency/optimal) and the four VDR genes after adjusting for other clinical parameters including, age, gender, dietary intake, physical activity, sun exposure, smoking and body mass index. The vitamin D status (deficiency vs. insufficiency/optimal) is based on biochemical normative values accepted by many international societies [vitamin D deficiency (<20 ng/mL), insufficiency (20–32 ng/mL) and optimal concentrations (>32 ng/mL)] [17].

## 3. Results

### 3.1. Baseline Charachteristics

Overall, 277 participants with a mean (±SD) age of 41 ± 12, 204 (74%) being female, were included in the study. Among the 277 subjects recruited, 46 (17%) had type 2 diabetes and 41 (15%) had hypertension. Using WHO cut-of-points for BMI, 65 (24%) subjects had normal BMI, 93 (34%) were overweight and 108 (39%) were obese at baseline (Table 1).

### 3.2. Distribution of four VDR Gene Polymorphisms in Group of 277 from Emirati Population

Figure 1 shows genotypes and allele percentage frequency distribution of four VDR gene polymorphisms in a group of 277 from the Emirati population. The frequencies of VDR genotypes for Bsml are GG = 37%, AG = 44%, and AA = 20.7%.

### 3.3. Vitamin D Concentrations According to Distribution and Presence or Absence of Genotypes and Alleles of Four VDR Gene Polymorphisms

Table 2 shows vitamin D concentrations according to genotype distribution of the four VDR gene polymorphisms. There were statistically significant differences in vitamin concentrations between different genotypes of the four VDR gene polymorphisms (*p* < 0.05). Table 3 shows vitamin D concentrations according to the presence or absence of genotypes and allele distribution of the four VDR gene polymorphisms. There was a statistically significant difference in vitamin concentrations between subjects with and without genotypes AA and AG and allele G in Apal VDR gene polymorphisms only (*p* < 0.05).

### 3.4. Genotype Distribution Based on Gender, BMI and Diabetes, Hypertension

Multiple logistic regression analysis revealed significant and independent association between vitamin D status and age and sex only (*p* < 0.05), (Table 4). We found no statistically significant independent associations between the VDR gene polymorphisms assessed and vitamin D status (deficiency vs. insufficiency/optimal) after adjusting for clinical parameters known to influence vitamin D status including age, sex, dietary intakes, physical activity, sun exposure, smoking and body mass index of study population (Table 4).

Figure 2, Figure 3 and Figure 4 show the genotypes and allele percentage frequency distribution of all four VDR gene polymorphisms, Bsml, Taql, Apal and Fok1 among subjects with and without diabetes, hypertension and obesity. There were no statistically significant differences in the frequency of the genotypes and alleles of Bsml, Taql, Apal and Fok1 VDR genes among patients with diabetes and hypertension compared to those without these medical conditions except in Fok1 AA genotype and G allele (Figure 2 and Figure 3). No statistically significant differences were found in the frequency of the genotypes and alleles of Bsml, Taql, Apal and Fok1 VDR genes in obese and overweight subjects compared to normal weight subjects (Figure 4). Furthermore, no statistically significant difference was found in BMI between subjects with and without the genotype and alleles of all four VDR genes.

## 4. Discussion

We examined the prevalence of the common four single-nucleotide polymorphisms in the VDR gene including BsmI, FokI, TaqI and ApaI in a large sample of the UAE population and compared vitamin D concentrations between subjects with and without the genotypes and alleles of all four VDR genes. In addition, we compared their association with chronic diseases common in UAE society, including diabetes, hypertension and obesity. Although we found statistically significant differences in vitamin concentrations between different genotypes of the four VDR gene polymorphisms, no association was found after adjusting clinical parameters known to influence vitamin D status. We also found no association with chronic diseases known to be prevalent in UAE society, such as obesity and its associated pathologies including diabetes and hypertensin, except in the Fok1 AA genotype and G allele.

### 4.1. Genotypes and Allele Percentage Frequency Distribution Compared to Previous Studies

The prevalence of some of the genotypes and alleles is in agreement with previous studies from the UAE community and other societies. For example, the frequency distribution of Bsml is in agreement with a previous study from the UAE (Saadi et al. findings: GG = 37%, AG = 42%, AA = 20%; our findings: GG = 37%, AG = 44%, AA = 20.7%) [17]. For the Taql VDR genotype, our result agree with another study’s findings from the UAE and a study from France [15,24]. Although not many studies in the UAE have reported the frequency distribution of the SNP3 Apal, a study from Poland has reported the relationship between SNP3 Apal, vitamin D and the occurrence of cardiovascular disease (CVS) [25]. Our results of SNP4 Fok1 genotypes and allele percentage frequency distribution are different from previous studies, including one from the UAE [15]. A study from India of the frequency of the Fok I and Taq I variants in healthy Indian individuals and its association with 25-OH-Vitamin D levels reported that the distribution of the polymorphic loci Fok I and Taq I vary considerably not only in different populations, but also within India [8]. Overall, the reasons for the difference in vitamin D genotypes and allele nucleotide polymorphisms percentage frequency distribution appear to be differences in sample size and sampling methods used between different studies. The unrepresentative or biased samples made it difficult to make valid generalizations about vitamin D genotypes and allele percentage frequency distribution within and between different populations. 

### 4.2. Vitamin D Concentrations According to the Presence or Absence of Genotypes and Allele Distribution of Four VDR Gene Polymorphism

Although we found statistically significant differences in vitamin concentrations between different genotypes of the four VDR gene polymorphisms, no statistically significant independent associations were found between the four VDR gene polymorphisms assessed and D status after adjusting for clinical parameters known to influence vitamin D status, such as dietary intake, physical activity, sun exposure, smoking and body mass index. Another previous study from the UAE found no association between vitamin D levels and bone turnover markers and Bsml VDR gene polymorphism [17]. In contrast, a study from India reported a significant association between vitamin D levels and Taq 1 SNP but not with the Fok I [8]. 

### 4.3. The Prevalence of Vitamin D VDR Receptor Genotypes and Alleles in Subjects with Obesity, Diabetes Mellitus and Hypertension Compared to Those without These Conditions 

Circulating vitamin D levels, action and underlying vitamin D receptor VDR genetic polymorphisms have been linked to many common diseases such as obesity and associated pathologies, including diabetes, hypertension and CVS diseases. This is important because of the co-existence of pathologically high proportions of obesity and vitamin D deficiency in our population. Many vitamin D genetic and alleleic polymorphisms have been identified, and their effects on VDR protein function and consequently vitamin D levels and health impact have been studied. The BsmI, FokI, TaqI and ApaI polymorphisms in the VDR gene have been the most studied. In this study we found no statistically significant differences in the percentage frequency distribution of all four VDR gene polymorphisms, Bsml, Taql, Apal and Fok1, among subjects with obesity, diabetes mellitus and hypertension compared to those without these conditions, except in the Fok1 AA genotype and G allele in subjects with diabetes and hypertension compared with those without these conditions. The association between vitamin D receptor VDR polymorphism and the risk of chronic diseases including obesity, hypertension, diabetes and other cardiovascular diseases therefore remains unclear. A recent preliminary study assessed the relationship between the VDR genotypes, plasma concentrations of vitamin D metabolites, and the occurrence of cardiovascular and metabolic disorders in 58 Polish patients treated for various cardiological diseases. Among patients with the TT genotype, frequency of hypertension was higher than among carriers of other ApaI genotypes (*p* < 0.01). In addition, carriers of the TT ApaI, TC TaqI, and GA BsmI genotypes had an increased risk of obesity, while the presence of the FokI TT genotype was associated with a higher incidence of heart failure and hypertension. In conclusion, this study reported that the BsmI AA genotype can be protective against CVS, but this observation needs to be confirmed on a larger group of patients. Particular VDR genotypes were associated with 25-hydroxyvitamin-D levels, and the mechanism of this association should be further investigated [22]. Another cross-sectional study that investigated relationships of vitamin D receptor gene VDR polymorphisms to the components of metabolic syndrome (MetS) including obesity, hypertension, diabetes and dyslipidemia among 198 (148 females, 50 males) Arab adults residing in the United Arab Emirates reported that VDR gene polymorphisms were not associated with MetS. However, the authors reported that it may affect the severity of some of components of MetS, namely the association of BsmI with obesity, FokI and BsmI with dyslipidemia and FokI with systolic blood pressure [26]. Another cross-sectional study from the UAE investigated the association between VDR polymorphisms and type 2 diabetes (T2DM) among 264 patients with T2DM, and 91 healthy controls were enrolled. The TaqI variant was shown to be associated with high cholesterol and LDL-cholesterol levels in T2DM patients, while BsmI was associated with lower BMI and lower LDL cholesterol levels. Their results implied that alleles and haplotypes of the VDR gene are associated with the susceptibility to T2DM in the Emirati population [27]. A meta-analysis of 14 studies of the association of the four polymorphisms FokI, BsmI, ApaI and TaqI in the VDR gene with the susceptibility to T2DM reported no significant associations between BsmI, ApaI and TaqI variants and T2DM risk. The report did suggest an increased risk of T2DM in subjects with the FokI VDR gene polymorphism in an Asian population [28]. The association of low vitamin D level and its receptor polymorphism with obesity was studied in 300 Saudi men. The report suggested that both bb of BsmI and tt of TaqI genotypes were higher in the obese group compared with the lean group and that low vitamin D levels and VDR BsmI and Taq1 genotypes may be a risk factor of obesity [29]. A study from Iran which looked at the association of the VDR gene ApaI, BsmI, and TaqI polymorphisms with obesity in 348 obese and 320 non-obese subjects reported an increased risk of obesity in subjects with VDR ApaI polymorphism. In particular, the A allele and the AA genotype in ApaI were associated with the obesity phenotypes [30]. Another systematic review of the association between genetic polymorphisms and obesity in the Arab world included 59 studies with a total of 15,488 cases and 9760 controls in the final analysis. A total of 76 variants located within or near 49 genes were reported to be significantly associated with obesity. Among the 76 variants, two were described as unique to Arabs, as they have not been previously reported in other populations, and 19 were reported to be distinctively associated with obesity in Arabs but not in non-Arab populations. The authors concluded that there appears to be a unique genetic and clinical susceptibility profile of obesity in Arab patients [31]. 

### 4.4. Strengths and Weakness

Our study is one of the very few studies to measure the four single-nucleotide polymorphisms in the VDR gene including BsmI, FokI, TaqI and ApaI in a large sample of the UAE population with a high prevalence of overweight and obesity (76%), diabetes (17%) and hypertension (15%). We used BMI as a measure of adiposity in our study. Although BMI is an easily accessible measure of excessive body weight in the general population, it cannot differentiate between fat and lean body mass. Individuals with normal weight can still demonstrate harmful adiposity traits [32]. Nevertheless, and similar to most previous studies reported on the relationship between vitamin D VDR genetic polymorphisms and risk or severity of chronic diseases, our sample selection and size may not be representative of the whole UAE population. Studies of the relationships between VDR gene polymorphisms and obesity-related diseases need to be powerful enough to detect a statistically significant association if one exists. Most previous studies reported in this area, however, have major heterogeneity and weaknesses in relation to study design, sample size, variables measured and control for confounders. A meta-analysis of these multiple studies may also have a role in clarifying the association. In this study, we also made an attempt to identify a potential independent effect of the four single-nucleotide polymorphisms on vitamin D status by adjusting for a number of clinical parameters likely to affect vitamin D status. 

## 5. Conclusions

Although we found statistically significant differences in vitamin concentrations between different genotypes of the four VDR gene polymorphisms, no independent associations were found between VDR gene polymorphisms assessed and D status (deficiency vs. insufficiency/optimal) after adjusting for clinical parameters known to influence vitamin D status. We also found no association of vitamin D receptor gene polymorphism SNPs with chronic diseases known to be prevalent in UAE society, such as obesity, diabetes and hypertension. There is still, however, the need for large well-designed epidemiological studies to evaluate the relationships between VDR gene polymorphisms and obesity-related diseases in an appropriate representative sample of the target population to which the findings will be referred. This is because of the co-existence of pathologically high proportions of obesity and vitamin D deficiency conditions in our society and the Middle East at large.

## Figures and Tables

**Figure 1 biomedicines-11-01202-f001:**
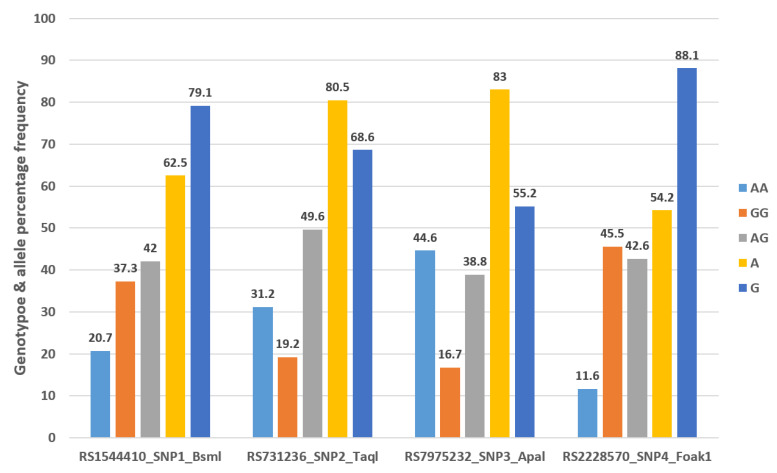
Genotype and allele percentage frequency distribution of four VDR gene polymorphisms in the Emirati population.

**Figure 2 biomedicines-11-01202-f002:**
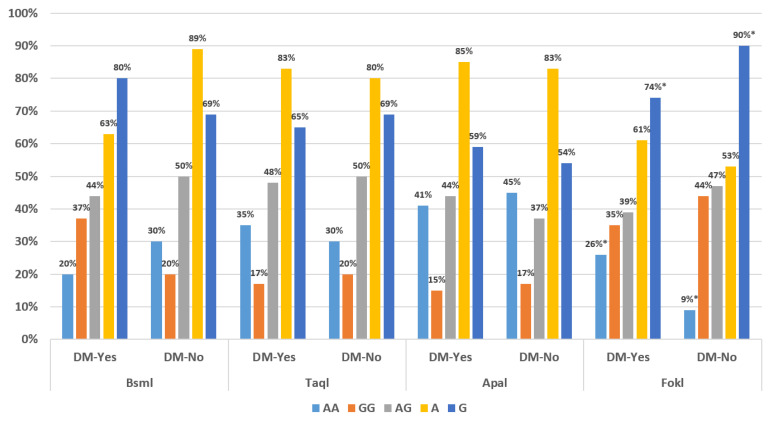
Genotypes and allele percentage distribution of four VDR gene polymorphisms in subject with diabetes (DM-yes) compared with those without diabetes (DM-No). * *p* < 0.05.

**Figure 3 biomedicines-11-01202-f003:**
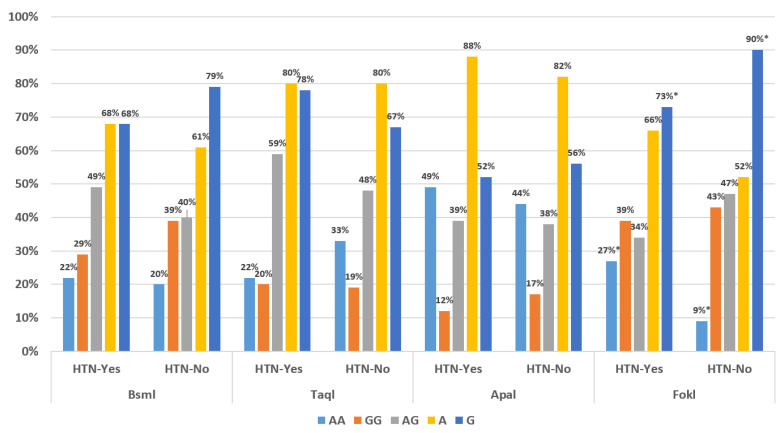
Genotypes and allele percentage distribution of four VDR gene polymorphisms in subject with hypertension (HTN-yes) compared with those without hypertension (HTN-No). * *p* < 0.05.

**Figure 4 biomedicines-11-01202-f004:**
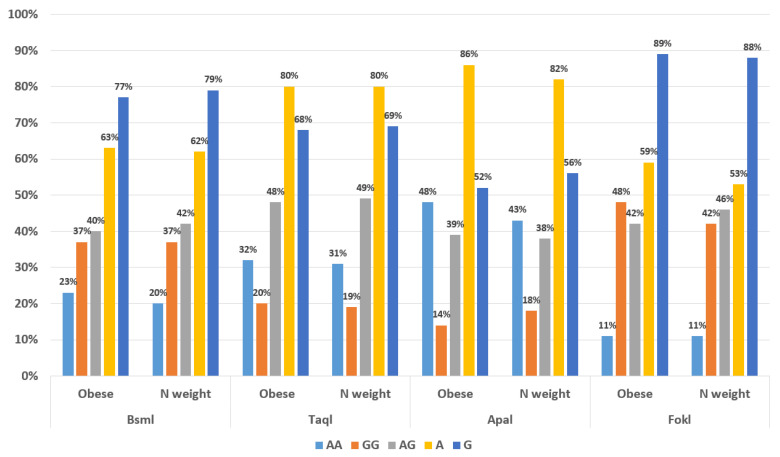
Genotypes and allele percentage distribution of four VDR gene polymorphisms in obese subjects compared to normal weight (N weight) subjects.

**Table 1 biomedicines-11-01202-t001:** Baseline clinical, anthropometric and metabolic characteristics of the study population.

Variables	Baseline Clinical, Anthropometric and Metabolic Characteristics of the Study Population	Mean (SD) Unless Otherwise Stated (n = 277)
Age (years)		41 (12)
Sex, female *n* (%)		204 (74)
Smoking *n* (%)		
	Current	35 (13)
	Ex-smoker	12 (4)
	Never smoked	225 (81)
Body mass index (BMI) *n* (%)		
	Normal weight (BMI 18.5–25)	65 (24)
	Overweight (BMI 25.1–29.9)	93 (34)
	Obese (BMI ≥ 30)	108 (39)
Physical activity *n* (%)		
	Not active	32 (12)
	Moderately active	146 (53)
	Very active	96 (35)
Diabetes *n* (%)		46 (17)
Hypertension *n* (%)		41 (15)
Hs CRP (mg/L) *		3.5 mg (3)
Glucose (mmol/L) *		6.3 (2.5)
Total cholesterol (mmol/L) *		4.9 (0.9)
Urea (mmol/L) *		4.1 (1.5)

* All values fall within normal limits.

**Table 2 biomedicines-11-01202-t002:** Vitamin D concentrations (ng/mL) according to genotypes distribution of four VDR gene polymorphisms.

	AA	GG	AG	*p* Value
RS1544410_SNP1_Bsml	24.5 (11)	22.1 (11)	23.9 (10)	0.046
RS731236_SNP2_Taql	22.0 (11)	24.0 (11)	23.9 (10)	0.054
RS7975232_SNP3_Apal	25.0 (11)	22.3 (11)	21.7 (9)	0.008
RS2228570_SNP4_Foak1	24.0 (10)	23.9 (12)	22.6 (9)	0.007

**Table 3 biomedicines-11-01202-t003:** Mean (SD) vitamin D concentrations (ng/mL) according to presence or absence of genotypes and allele distribution of four VDR gene polymorphisms.

	AA		GG		AG		A		G	
	**Yes**	**No**	**Yes**	**No**	**Yes**	**No**	**Yes**	**No**	**Yes**	**No**
RS1544410_SNP1_Bsml	25.5 (11)	23.2 (10)	22.1 (11)	24.2 (10)	23.9 (10)	23.1 (11)	23.9 (10)	22.7 (12)	23.0 (10)	25.0 (12)
RS731236_SNP2_Taql	22.0 (11)	24.1 (10)	24.0 (11)	23.3 (11)	23.9 (10)	23.0 (11)	23.2 (11)	24.5 (11)	23.9 (11)	22.3 (11)
RS7975232_SNP3_Apal	25.1 (11)	22.1 * (10)	22.3 (11)	23.7 (11)	21.7 (9)	24.5 * (11)	23.6 (10)	22.9 (12)	21.9 (10)	25.3 * (12)
RS2228570_SNP4_Foak1	23.9 (10)	23.4 (11)	22.6 (9)	24.1 (11)	23.9 (12)	23.0 (10)	22.9 (10)	24.1 (12)	23.3 (11)	24.6 (11)

* *p* < 0.05 for difference between those with and without the genotypes and allele distribution of four VDR gene polymorphisms.

**Table 4 biomedicines-11-01202-t004:** Multiple logistic regression analysis of the influence of four VDR gene polymorphisms and some clinical prognostic variables on vitamin D status (deficiency vs. insufficiency/optimal) of study population.

	Standardized Coefficients B	Standard Error	*p* Value	Exp (B)	95.0% Confidence Interval for B	
					Lower Bound	Upper Bound
Age (years)	0.018	0.008	0.022	1.018	1.003	1.034
Gender (male/female)	−0.763	0.241	0.002	0.466	0.291	0.748
BMI	−0.006	0.008	0.443	0.994	0.978	1.010
Sun exposure	0.009	0.044	0.838	1.009	0.926	1.099
Diet	0.010	0.036	0.782	1.010	0.941	1.083
Physical activity	0.050	0.154	0.748	1.051	0.777	1.422
Smoking	0.071	0.142	0.616	1.074	0.813	1.417
RS1544410_SNP1_Bsm	0.179	0.146	0.221	1.196	0.898	1.593
RS731236_SNP2_Taql	−0.067	0.127	0.598	0.935	0.730	1.199
RS7975232_SNP3_Apal	−0.079	0.064	0.217	0.924	0.815	1.047
RS2228570_SNP4_	−0.156	0.144	0.279	0.855	0.645	1.135

## Data Availability

Data is available upon request to the corresponding author.

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
