# Peer review of "Frequency of Vitamin D Receptor Gene Polymorphisms in a Population with a very High Prevalence of Vitamin D Deficiency, Obesity, Diabetes and Hypertension"

_biomedicines, 2023, doi:10.3390/biomedicines11041202_

Round 1

Reviewer 1 Report

The paper by Salah Gariballa et al. describes the results of the clinical study in which vitamin D receptor (VDR) gene polymorphisms in the Emirati population with a high prevalence of vitamin D deficiency, obesity, diabetes and hypertension, was studied. The authors found statistically significant differences in vitamin concentrations between four different genotypes of VDR gene polymorphisms, but no association after adjusting for clinical parameters known to influence vitamin D status.

The authors mention that previously published data about correlations between VDR polymorphic variants and various diseases were conflicting. This is why they wanted to study this issue in Emirati population. Unfortunately, there is an inherent problem connected with studies which describe correlations between gene polymorphisms and diseases. The statistical significance value of p<0.05 which is used in such studies provides that one in 20 studies will give false positive results. Since authors are more willing to publish positive, than negative results in their papers, the proportion of false positive results in published papers is even higher.

Therefore, in my opinion, the data about polymorphisms and diseases should be published only in these cases, when real mechanism of the polymorphism influencing the origin or course of the disease can be proven.

For this study, I would suggest just publication of distribution of the polymorphisms in Emirati population in a less prestigious journal.

Author Response

Reviewer 1 

  1. Therefore, in my opinion, the data about polymorphisms and diseases should be published only in these cases, when real mechanism of the polymorphism influencing the origin or course of the disease can be proven.

As we have already stated in the introduction a number of loss-of-function mutations of the VDR gene linked to diseases have been identified.  And, because of the large size of the VDR gene more SNPs are expected to identified in future.  In addition, the relationship between vitamin D and its VDR receptor polymorphism with obesity and related pathologies have also been described.  Both obesity and vitamin D deficiency are highly prevalent in UAE citizens, however their adverse and additive health implications are not yet clear.  Mutations of the VDR gene and allelic polymorphisms may also influence disease susceptibility and responses to levels of circulating vitamin D especially in areas where D deficiency is highly prevalent, hence the need for our study and more research in the field.

Reviewer 2 Report

In this paper the Authors aim to assess the Frequency of Vitamin D receptor gene polymorphisms in a population with a very high prevalence of vitamin D deficiency, obesity, diabetes and hypertension.. It is an interesting and debated topic. A comprehensive and extensive literature review of the NCBI database PubMed was also carried out. The article was well conducted and it is interesting in its fields. It is a well-structured paper, written in good English and the References are up dated. 

Minor issues:

In the “discussion” section I suggest to better analyze the drawbacksand vitamin deficits of the different bariatric procedures in obese patients. Therefore, the following paper should be considered:

“Lucido FS, Scognamiglio G, Nesta G, Del Genio G, Cristiano S, Pizza F, Tolone S, Brusciano L, Parisi S, Pagnotta S, Gambardella C. It is really time to retire laparoscopic gastric banding? Positive outcomes after long-term follow-up: the management is the key. Updates Surg. 2022 Apr;74(2):715-726. doi: 10.1007/s13304-021-01178-1. Epub 2021 Oct 1. PMID: 34599469; PMCID: PMC8995288.”

“Del Genio G, Tolone S, Gambardella C, Brusciano L, Volpe ML, Gualtieri G, Del Genio F, Docimo L. Sleeve Gastrectomy and Anterior Fundoplication (D-SLEEVE) Prevents Gastroesophageal Reflux in Symptomatic GERD. Obes Surg. 2020 May;30(5):1642-1652. doi: 10.1007/s11695-020-04427-1. PMID: 32146568.”

“Pizza F, Lucido FS, D'Antonio D, Tolone S, Gambardella C, Dell'Isola C, Docimo L, Marvaso A. Biliopancreatic Limb Length in One Anastomosis Gastric Bypass: Which Is the Best? Obes Surg. 2020 Oct;30(10):3685-3694. doi: 10.1007/s11695-020-04687-x. PMID: 32458362.”

Author Response

Reviewer 2

  1. Minor issues:In the “discussion” section I suggest to better analyze the drawbacksand vitamin deficits of the different bariatric procedures in obese patients. Therefore, the following paper should be considered:

“Lucido FS, Scognamiglio G, Nesta G, Del Genio G, Cristiano S, Pizza F, Tolone S, Brusciano L, Parisi S, Pagnotta S, Gambardella C. It is really time to retire laparoscopic gastric banding? Positive outcomes after long-term follow-up: the management is the key. Updates Surg. 2022 Apr;74(2):715-726. doi: 10.1007/s13304-021-01178-1. Epub 2021 Oct 1. PMID: 34599469; PMCID: PMC8995288.”

“Del Genio G, Tolone S, Gambardella C, Brusciano L, Volpe ML, Gualtieri G, Del Genio F, Docimo L. Sleeve Gastrectomy and Anterior Fundoplication (D-SLEEVE) Prevents

Gastroesophageal Reflux in Symptomatic GERD. Obes Surg. 2020 May;30(5):1642-1652. doi: 10.1007/s11695-020-04427-1. PMID: 32146568.”

“Pizza F, Lucido FS, D'Antonio D, Tolone S, Gambardella C, Dell'Isola C, Docimo L, Marvaso A. Biliopancreatic Limb Length in One Anastomosis Gastric Bypass: Which Is the Best? Obes Surg. 2020 Oct;30(10):3685-3694. doi: 10.1007/s11695-020-04687-x. PMID: 32458362.”

We have studied the above 3 references provided and found in none of them any relevance to our study main objectives, results or discussion.  All three 3 references about surgery technique , anthropometric or  surgical outcomes of obese patients underwent  different bariatric surgeries, 

For example in reference number 1  “Lucido FS, Scognamiglio G, Nesta G, Del Genio G, Cristiano S, Pizza F, Tolone S, Brusciano L, Parisi S, Pagnotta S, Gambardella C. It is really time to retire laparoscopic gastric banding? Positive outcomes after long-term follow-up: the management is the key. Updates Surg. 2022 Apr;74(2):715-726. doi: 10.1007/s13304-021-01178-1. Epub 2021 Oct 1. PMID: 34599469; PMCID: PMC8995288.” Not a single time mention of the words; vitamin D or its receptor VDR polymorphism

Round 2

Reviewer 1 Report

My opinion is the same as before.

Author Response

Ms. Roxana Pluteanu

Assistant Editor

Biomedicines Journal

3 Aprl 2023 

Dear Ms. Roxana Pluteanu

Manuscript ID: biomedicines-2287817 - Minor Revisions

Title: Frequency of Vitamin D receptor gene polymorphisms in a population with a very high prevalence of vitamin D deficiency, obesity, diabetes and hypertension.

Thank you for your email of the 30th March 2023 and for the helpful comments from yourself and the reviewers.  Although no specific points raised by reviewer 1 to address we have been guided by the tick box points feedback on improving the background, design and study limitations and conclusions

Point by point response:

Reviewer 1  

  1. Study back ground must be improved

New text with 3 new references provided on the links between low vitamin D, obesity and related diabetes hence the importance of studying the relationship between vitamin D and its VDR receptor polymorphism with obesity and related pathologies given the influence of the VDR receptor polymorphism on vitamin D levels and functions (lines 55-61 and lines 402-408)

  1. Research design

Study population details edited (lines 100-106) and a new text provided on statistical power of the study based on our sample size (lines 146-152)

  1. Limitations and conclusions

We have added new text on study limitations acknowledging the fact that studies of the relationships between VDR gene polymorphisms and obesity related diseases needs to be powerful enough to detect a statistically significant association if one exists.  But equally important we also added that a meta-analysis of multiple small studies may also have a role in clarifying the association (lines 341-347).

New text also added to the conclusion (lines 357-8)

We trust that you will now find the manuscript suitable for publication in Biomedicine journal 

Yours sincerely

Dr SE Gariballa, Corresponding Author